# Child and Adult Care Food Program: Family Childcare Home Providers’ Perceptions of Impacts of Increased Meal and Snack Reimbursement Rates during the COVID-19 Pandemic

**DOI:** 10.3390/nu16193241

**Published:** 2024-09-25

**Authors:** Lorrene D. Ritchie, Kassandra A. Bacon, Celeste Felix, Danielle L. Lee, Samantha Kay-Daleiden Marshall, Elyse Homel Vitale, Susana L. Matias

**Affiliations:** 1Nutrition Policy Institute, Agriculture and Natural Resources, University of California, 1111 Franklin Street, Eleventh Floor, Oakland, CA 94607, USA; kabacon@ucanr.edu (K.A.B.); cmfelix@ucanr.edu (C.F.); dnilee@ucanr.edu (D.L.L.); 2CACFP Roundtable, San Diego, CA 92172, USA; samantha@cacfproundtable.org (S.K.-D.M.); elyse@cacfproundtable.org (E.H.V.); 3Department of Nutritional Sciences and Toxicology, University of California, Berkeley, 225 Morgan Hall, Berkeley, CA 94720, USA; slmatias@berkeley.edu

**Keywords:** child and adult care food program, nutrition policy, childcare, reimbursement, preschool-age children

## Abstract

**Introduction:** The U.S. Child and Adult Care Food Program (CACFP) provides tiered reimbursements for healthy foods for children at participating family childcare homes (FCCH). Higher tier 1 reimbursements are for providers who operate in low-income communities or who are themselves living on a low income. All FCCHs received a higher rate to address food insecurity during the COVID-19 pandemic. **Methods:** A survey was administered in the spring of 2023 to a randomly selected sample of licensed California FCCHs to assess the perceived impacts of the increased reimbursement on CACFP participation and anticipated challenges with reinstated tiered rates. A total of 518 surveys (261 tier 1, 257 tier 2) were analyzed using linear or logistic regression, adjusting for confounders. **Results:** Among tier 1 and tier 2 providers combined, over half reported lowering out-of-pocket spending for food (59%) and serving greater variety (55%) and quality (54%) of foods. Tier 2 providers reported experiencing more benefits (*p* < 0.05) and tended to be more likely to implement optional CACFP best practices (although not significantly different between tiers). Most FCCH providers found reimbursement rates were inadequate before (83%) the pandemic; this amount decreased to 54% post-pandemic for tier 1 and tier 2 providers combined. **Conclusions:** The temporary CACFP reimbursement positively impacted the perceived quality and variety of foods served to children, especially among tier 2 providers. Increased reimbursements for all FCCHs may ensure children have access to the healthy meals and snacks provided by the CACFP.

## 1. Introduction

In the U.S., licensed childcare represents the largest institutional setting for ensuring children’s access to nutritious foods during the critical early years of development. Licensed childcare can range from relatively large centers or preschools with multiple staff taking care of classrooms of children to family childcare homes (FCCH) operated by a caretaker in her own home caring for a smaller number of children by herself or with a small number of staff. While licensing requirements vary by state and by childcare type, they generally include initial and ongoing monitoring of provider training and immunizations and child safety. Within childcare, the federal Child and Adult Care Food Program (CACFP) provides meals and snacks to over 4 million children [1,2]. The CACFP provides licensed childcare centers and family childcare homes that meet eligibility criteria and adhere to nutrition standards reimbursements for up to two meals and one snack per child each day [3,4].

Research has demonstrated that the CACFP’s combination of nutrition requirements and financial support enables participating providers to offer healthier food options than those that do not participate in the program. This is particularly important as children may consume up to two-thirds of their daily food intake while at childcare [5,6]. Compared to non-participants, CACFP participants serve more vegetables [7,8,9,10,11], more whole grains and less refined grains [11,12,13], healthier beverages [10,13,14,15], less sweet- and snack-type foods [8,11,13], and less trans and saturated fats [15]. Additionally, although less well-studied, the CACFP may help reduce child food insecurity [16,17,18].

The CACFP also plays a significant role in addressing child health disparities. Non-Hispanic Black and Hispanic children and children living in households with low income are the most at risk of experiencing health-related conditions such as food insecurity, poor diet quality, and obesity compared to non-Hispanic White children and children living in households with higher income [19,20,21]. The program’s meals and snacks, at no or reduced cost to families, can also contribute to family financial well-being. In California, approximately six in ten children reached by the CACFP are non-White and are more likely than their White counterparts to live in poverty [22,23]. FCCHs provide cost-effective and convenient care for children of various ages in a home setting (often located close to families and offering longer, more flexible hours of care, such as evenings and weekends), thereby catering to families with low income more than centers [24].

However, compared to childcare centers, few nutrition standards apply to FCCHs outside the CACFP [25]. Further, children in FCCHs may face a higher risk of obesity than children cared for in their own homes or in childcare centers [26]. Differences in the CACFP administration between FCCHs and centers also may affect the program’s reach to families with low income. While the CACFP reimburses centers based on whether children qualify for free, reduced-price, or paid meals (like the U.S. national school meal programs), FCCHs receive tiered reimbursements: higher tier 1 rates apply to FCCHs in low-income neighborhoods or operated by providers with low income, while lower tier 2 rates apply to all others. Although these tiers were designed to direct more resources to families with low income, all FCCHs must follow the same requirements, and this structure introduces administrative burdens [27,28], potentially limiting CACFP access by young children. 

The U.S. Department of Agriculture (USDA) has rules and provides guidance for rule implementation to state agencies who administer CACFP. State agencies may add more stringent rules, and the implementation of the federal rules varies across the nation. Differences may include not allowing certain types or brands of food to be served, how to correctly document infant formula on the menus, or whether electronic forms and signatures are allowed. Additionally, all FCCHs must go through a third party, non-profit or public, sponsoring organization to enroll in the CACFP and to process paperwork and receive reimbursements.

Despite the benefits, not all FCCH providers choose to participate in the program. Nationally, a decline of FCCH participation in the CACFP began in 1997 when tiers were instituted [27]. In California, from 2010 to 2020, the number of CACFP lunches served by FCCHs decreased by 21%, resulting in over 20,500 fewer children served daily [22]. The COVID-19 pandemic further exacerbated this decline, with a 21% drop in FCCH reimbursements from November 2019 to November 2020 [29]. In 2019, only two-thirds of eligible FCCHs nationally participated in the program [30]. Given that one in five children in the U.S. will receive care at an FCCH before kindergarten [31], these trends indicate a potential gap in food and nutrition security among young children.

In response to these systemic challenges and the increase in food insecurity during the COVID-19 pandemic, the USDA temporarily waived the tiered reimbursements from July 2021 to July 2023, allowing all FCCHs—regardless of tier—to access the same higher meal and snack reimbursement rates [32]. For example, from July 2021 to July 2022, in most states, the federal per child daily reimbursements for breakfast, lunch, and a snack increased from $4.78 for tier 1 and $2.29 for tier 2, to $5.67 for both tiers. This temporary policy change, which ended in July 2023, provided a unique opportunity for a natural experiment. Our study aimed to answer the following research questions: what are the perceived impacts of the increased CACFP reimbursements on the benefits of participating in the program, and what are the potential challenges anticipated with reinstating the tiered reimbursement rates as reported by FCCH providers in the state of California?

## 2. Methods

### 2.1. Design and Sample Recruitment

In the spring of 2023, 2000 licensed California FCCHs who participated in the CACFP (half tier 1; half tier 2) were randomly selected from a total of approximately 28,000 FCCHs in state administrative data. Links and QR codes to access the survey (English and Spanish) via computer or mobile device were sent by email and postcard to each selected FCCH provider. The initial survey invitation was sent by an entity known to providers, the California Department of Social Services, which is the state administrator of the CACFP. In addition, the CACFP Roundtable, a national CACFP advocacy and support organization, encouraged participation in the survey via email communications to California members, and all FCCH sponsoring organizations in the state were sent sample communications about the survey to share with their FCCHs. Up to three follow-up emails and one follow-up postcard were sent to FCCHs who did not complete the survey. A printed copy was mailed, and follow-up phone calls were made to providers who did not complete the survey online. Providers were offered the opportunity to enter a raffle for one of ten gift cards valued at $100 each to incentivize survey participation. The University of California, Davis, and State of California Institutional Review Boards reviewed and approved this study.

### 2.2. Measures

Survey questions were adapted from instruments used by this research team in previous statewide surveys on childcare nutrition ([8,13,14,25] for questions on provider and FCCH operations), as well as in studies by others ([9,10,27,33,34,35,36,37,38,39,40,41] for questions on the CACFP). To ensure relevance and face validity, questions were reviewed and informed by a project advisory board, which included English and Spanish-speaking state and national experts on the CACFP, FCCH providers, and families who attend FCCHs. The survey was then pilot tested with advisory board members. The final survey was translated into Spanish and then checked by a native Spanish speaker before being programmed into Qualtrics (Qualtrics, 2023, Provo, UT, USA). The survey was designed to be completed using a mobile device (e.g., smartphone) in no more than 30 min to maximize equitable access.

The following measures were examined: perceived benefits experienced since receiving the higher reimbursements, anticipated changes following the end of the higher reimbursements, the perceived impact of the temporary increase in reimbursement on the nutritional quality of meals/snacks, the perceived ability to comply with CACFP’s optional best practices [42], and the estimated cost of preparing meals/snacks for children relative to the pre-COVID and higher COVID-19 reimbursement rates. The relevant survey questions and response options are provided as Appendix A.

### 2.3. Analysis

The data from paper surveys were entered into Qualtrics and then merged with the online survey data. Respondents must have answered at least one demographic question (age, gender, race/ethnicity, education, income or household size) and one additional survey question to be included in the analytic sample. An analytic sample of 518 was reached after removing 328 respondents who did not answer at least one demographic question (*n* = 143) and one additional survey question (*n* = 185). Descriptive statistics (means, SD and *n*, %) were used to describe the sample. To evaluate differences in survey responses by childcare category (tier 1 or tier 2), linear regression was used for continuous variables and logistic regression for categorical variables. Family childcare home characteristics were included in the analysis as covariates if there was a statistical difference with a *p* < 0.20 between tiers or were associated with the outcomes. The following characteristics were included as covariates: number of years as a licensed family childcare home, number of years participating in the CACFP, number of children in care, number of meals/snacks provided daily, type of childcare offered (half-day, full-day, evening, overnight, weekend), provider race/ethnicity, primary language (English or Spanish), number of training certificates completed regarding child development out of a total of 5 provided as options, highest level of education, and household income [43]. Analyses were conducted using R (software version 4.4.1).

## 3. Results

A total of 518 FCCHs (*n* = 261 tier 1, *n* = 257 tier 2) were included in the analysis. Characteristics of survey respondents by tier are provided in Table 1. FCCH providers in the total sample (tiers 1 and 2 combined) averaged 51 years of age. Most FCCH providers in the total sample were female (98%), used English as their primary language (86%), were Hispanic (38%) or non-Hispanic White (28%), had a household income between 100 and 200% of the federal poverty level (FPL; 62%), lived in a food secure household (80%), and completed at least some college (46%). Over half of the FCCH providers completed at least one training certification (56%), and three-quarters had one or more training certificates (75%). More than half of the total sample participated in the CACFP for 5 or more years (63%). Most FCCHs provided full-day care (99%) and served breakfast (91%), lunch (98%), and a morning or afternoon snack (85% and 94%, respectively). On average, FCCH providers had two employees (including themselves) and nine children in their care. Significant differences (*p* < 0.001) were observed between tier 1 and tier 2 FCCHs on the following characteristics: primary language spoken, non-Hispanic Asian/Pacific Islander, non-Hispanic White, high school graduate or less than high school, Bachelor’s degree or higher, household income less than 100% of the federal poverty level, number of certifications received, provide weekend, evening or overnight childcare, serve breakfast, dinner/supper and evening snack, care for toddlers (12–23 months), young children (2–5 years) and school aged children (6+ years), and serve children whose primary language is Spanish and Chinese. All of these variables were included as control variables in the analyses except for children’s primary language, as this is not likely associated with the outcomes.

Figure 1 shows the perceived benefits of receiving higher reimbursement rates during the COVID-19 pandemic. Over half of all FCCH providers surveyed (tiers 1 and 2 combined) reported lowering the amount they had to spend out-of-pocket for food costs (59%), serving a greater variety of foods (55%) and serving better quality (54%) foods. Slightly less than half of providers reported that they were better able to serve meals and snacks that children like (42%). Fewer providers reported that the higher reimbursements enabled them to not charge families more for childcare (28%) or lowered the food costs passed on to parents/guardians (14%). A minority of providers said that the higher reimbursement allowed them to increase the number of meals and snacks served (18%). Few (8%) reported no advantages from the higher reimbursements. After adjusting for potential confounding variables, significantly more tier 2 than tier 1 providers reported experiencing benefits from the higher reimbursement (with *p*-values ranging from *p* < 0.05–*p* < 0.001) except for lowering out-of-pocket costs for food and increasing the number of meals/snacks served, while significantly more tier 1 than tier 2 providers reported not experiencing any benefits (*p* < 0.05).

When asked if the higher reimbursement rates made it easier or harder to implement the optional CACFP best practices (Figure 2), the majority of tier 1 and tier 2 providers combined responded that it was relatively easier to implement each of the seven best practices, such as the following: offering whole fruits more than fruit juice; offering only natural, low-fat/reduced fat cheese; serving dark green, red/orange, starchy, and other vegetables or legumes at least once a week; providing whole grain-rich grains at least twice a day; offering a fruit or vegetable at snack time; serving processed meats no more than once per week; and offering only lean meats, nuts, and legumes as protein sources. Tier 2 providers tended to report it was easier than tier 1 providers for all practices, although there were no statistically significant differences between tiers.

When asked about the likelihood of having to make changes once the tiered CACFP reimbursement rates returned, both tier 1 and 2 FCCH providers (~one-quarter or more) said it was extremely likely that they would do the following: increase the amount families are charged for childcare and decrease the variety of foods provided at no charge to families (Figure 3). Few providers said it was extremely likely they would leave the CACFP and require families to provide food for children (14% tier 1, 12% tier 2), leave the CACFP but continue to offer food for children (10% tier 1, 13% tier 2), or close their business (4% tier 1, 3% tier 2). The responses to changes anticipated with the return of tiering were not significantly different between tier 1 and 2 providers except that tier 2 providers were significantly more likely than tier 1 providers to anticipate having to decrease the variety of foods offered to children (*p* < 0.05).

Finally, when asked about the adequacy of CACFP reimbursement rates in terms of covering actual meal and snack costs, most estimated that amounts were not enough pre-COVID (83% of tiers 1 and 2 combined), while half reported they were inadequate after the COVID-related increases (54% of tiers 1 and 2 combined) (Table 2). Few providers (<1% of tiers 1 and 2) reported the rates were too much before or after the increase. The differences between tiers 1 and 2 were observed for the adequacy of the higher COVID-19 reimbursement rate not being enough or being just right: more tier 1 than tier 2 providers reported the increased amount as not enough and more tier 2 than tier 1 providers reported the increased amount as just right. For over a quarter of providers (27% of tiers 1 and 2 combined) the CACFP reimbursement needed to provide two meals and one snack each day per child aged three to five years that met CACFP nutrition standards and appealed to children ranged between $6 and $8. When the midpoint of each reimbursement rate range was used to calculate a mean amount, the amount estimated was $7.34 for tier 1 and $7.54 for tier 2.

## 4. Discussion

In this U.S. study, California FCCH providers participating in the CACFP reported that the temporary higher meal and snack reimbursements received following the COVID-19 pandemic allowed them to increase the variety, quality, and healthfulness of foods served to young children under their care. These perceived improvements were particularly evident among tier 2 providers, whose reimbursement more than doubled compared to a much smaller increase for tier 1 providers. Many FCCH providers expressed concern that the return of tiers may limit food quality and quantity, potentially necessitating higher childcare fees for families. While under half indicated that returning to tiered rates might lead them to withdraw from the CACFP (45% in the combined sample of tier 1 and 2 FCCHs) or end their childcare business (16% overall), such outcomes could further exacerbate the ongoing decline in the U.S. in the number of licensed FCCHs and the proportion that participate in CACFP [4]. Given that CACFP participants generally provide better nutrition than nonparticipants [7,8,9,10,11,13,14,15], further reductions in access to the CACFP for young children is concerning [47].

Similar to the findings from our California study, a national survey conducted in the U.S. in 2023 identified that the primary challenge faced by FCCH providers (*n* = 1443) in relation to the CACFP was that reimbursements did not cover food costs [48]. Inadequate funding was reported by half (49%) of providers formerly on the program and by an even larger proportion of current CACFP participants (65%); however, no comparison by tier was conducted in this national sample [48]. Another national study found that inadequate funding for non-food administrative costs and burdensome paperwork were also major barriers for childcare centers’ participation in CACFP [35]. A study of FCCHs in Massachusetts one year after the 2017 update to the CACFP nutrition standards [3] found that despite not fully meeting all of the updated requirements, food costs increased by $0.27 per lunch and $0.25 per snack [49]. Furthermore, a study conducted prior to the pandemic compared FCCHs receiving the higher tier 1 and lower tier 2 reimbursements. In this study in Washington state, higher reimbursements were positively associated with better nutritional quality of the foods served to young children [50], reinforcing the need for higher reimbursement rates.

Qualitative research also revealed that childcare providers in the U.S. do not believe reimbursement amounts are adequate to cover the costs of foods meeting the CACFP requirements [38,51,52,53]. In-depth interviews with childcare stakeholders (center and FCCH providers on and off the program and state administrators and sponsors) in Arizona and New York highlighted the need for additional funding to overcome challenges with CACFP participation [38]. The high cost of CACFP-eligible foods was similarly noted as a barrier to CACFP participation by FCCH providers interviewed in Michigan [51] and Iowa [53].

In our California FCCH study, inadequate CACFP reimbursements appear to be particularly problematic for tier 2 providers. Although tiering was instituted in 1997 to allocate more federal resources to families with low income [33], it has been recommended that the higher pandemic-related reimbursement rates be made permanent [38]. At the time of our study, both tier 1 and 2 FCCH providers received $5.67 per child to serve breakfast, lunch, and a snack per day. In 2023, when tiering returned, a tier 1 FCCH provider serving these meals received $5.70 per child compared to only $2.72 per child if tier 2 (amount adjusted for inflation) [54]. Yet, both tier 1 and 2 providers must comply with the same program requirements. Even the higher tier 1 reimbursement rate was deemed inadequate by tier 1 providers in our study. When asked for the amount needed, the mean response for both tier 1 and 2 providers combined was just over $7.00.

Comparing our findings to studies from other countries is challenging due to differences in childcare operations. For example, high-income countries vary in the extent of federal funding for subsidized childcare [55]. There are also differences in the establishment and implementation of nutrition standards in early care and education settings across countries [56,57,58]. Additionally, not all countries subsidize food in the same way as the CACFP does in the U.S. [59]. Although many countries provided temporary financial support to childcare to address closures and other challenges during the pandemic [60,61,62], we found no studies in other countries documenting the impacts of increased funding for the food provided by FCCHs.

This study has several limitations. While conducted in California where the population represents over 11% of the U.S. total population [63], study results may not be generalizable to other states. Although all FCCH providers nationally received higher CACFP reimbursements during the initial years of the COVID-19 pandemic, there may be differences between states in how the CACFP is perceived and/or administered, which could impact FCCH responses. While our study sample’s demographics (age, race/ethnicity, and gender) are similar to another comprehensive study of California’s early care and education workforce [41], a response rate of 26% may have introduced bias, making findings non-representative of the state’s FCCH population. The administrative data used to select our sample indicated a lower average licensing duration compared to our sample, suggesting an underrepresentation of newer FCCH providers. In addition, CACFP barriers likely exist at the family, sponsor, and state levels, the evaluation of which was beyond the scope of this study. Future research should explore the perceptions of these other CACFP stakeholders as well as FCCH responses after the reinstatement of tiers. Not included in this analysis is the rising cost of food, which likely further influenced FCCH providers’ perceptions of CACFP reimbursement adequacy. During the time of this study, U.S. Consumer Price Index food at home prices increased from 0.9% in 2019 to 5.0% in 2023, hitting a 11.4% peak in 2022 [64]. Other federal pandemic-related support through provisions such as the CARES Act of 2020 [65] and the American Rescue Plan of 2021 [66] may have also impacted FCCH providers. Another study limitation is that data on the impacts of the higher reimbursements were self-reported by providers. Future evaluations should include objective measures of meal and snack quality provided by FCCHs.

Study findings suggest that the temporary increase in CACFP reimbursement rates positively impacted the quality and healthfulness of meals and snacks provided to young children, especially by tier 2 FCCH providers. Future studies should investigate whether the reinstatement of tiered reimbursements leads to less participation by FCCHs in the CACFP. Despite the program’s many benefits, the CACFP was underutilized before the pandemic. Most young children in the U.S., regardless of family income, consume an unhealthy diet, with an average Healthy Eating Index-2020 score (a comprehensive indicator of diet quality) of 61 (out of 100) for young children ages 2–5 years [67]. Given the critical need to improve child nutrition and to address the increasing prevalence of nutrition-related chronic conditions, such as diabetes and heart disease, among adults [68], it is time to reconsider tiered CACFP reimbursements for FCCHs. Increasing the reimbursement rates for FCCHs may ensure that more children have access to the healthy meals and snacks provided by the CACFP.

## 5. Conclusions

The CACFP promotes the enhanced nutrition of meals and snacks served to young children, especially to those in families living on low income, by providing nutrition requirements and financial support to childcare providers. Despite the many benefits of receiving financial support, CACFP FCCH providers perceive the reimbursement rates to be inadequate, especially tier 2 providers. While the increased reimbursement rates due to the temporary COVID-19 waiver positively impacted perceptions of the quality of meals and snacks served, providers in both tiers desired a higher reimbursement rate to provide meals and snacks that meet CACFP standards and appeal to the children in their care. Increasing reimbursements for all FCCHs may support increased FCCH participation in the CACFP, ensuring more U.S. children benefit from the healthy meals and snacks provided by the CACFP.

## Figures and Tables

**Figure 1 nutrients-16-03241-f001:**
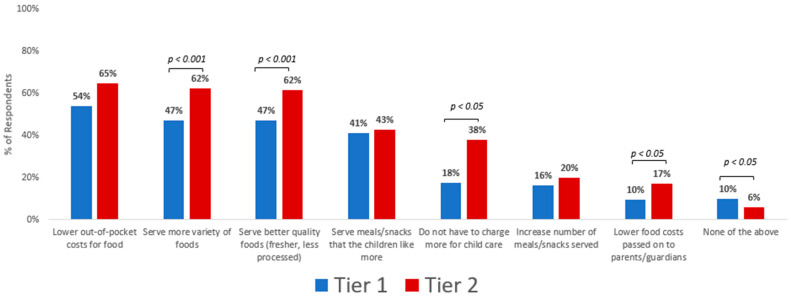
Perceived benefits reported with higher CACFP reimbursement by tier 1 (*n* = 261) and tier 2 (*n* = 257) family childcare home providers. Significant differences between tiers were determined using logistic regression adjusting for: years licensed, years participating in CACFP, number of children in care, type of childcare offered (full, half, weekend, evening, overnight), number of meals/snacks provided daily, provider race/ethnicity, primary language, education, number of certifications received, and household income).

**Figure 2 nutrients-16-03241-f002:**
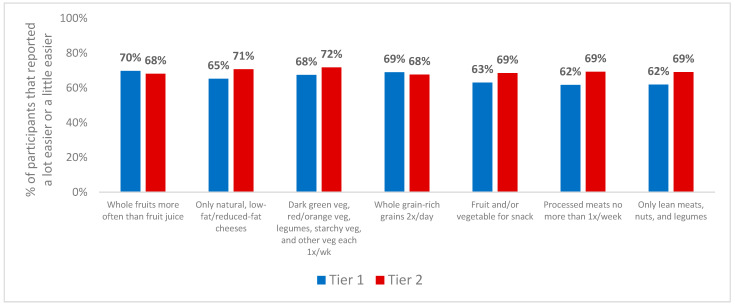
Perceived ease in implementing CACFP best practices for children by tier 1 (*n* = 261) and tier 2 (*n* = 257) family childcare home providers while receiving higher reimbursements. Response options ‘a little easier’ and ‘a lot easier’ were combined and response options ‘a little harder’ and ‘a lot harder’ were combined. Significant differences between tiers were determined using logistic regression adjusting for: years licensed, years participating in CACFP, number of children in care, type of childcare offered (full, half, weekend, evening, overnight), number of meals/snacks provided daily, provider race/ethnicity, primary language, education, number of certifications received, and household income. See Appendix A for frequencies of all responses by tier).

**Figure 3 nutrients-16-03241-f003:**
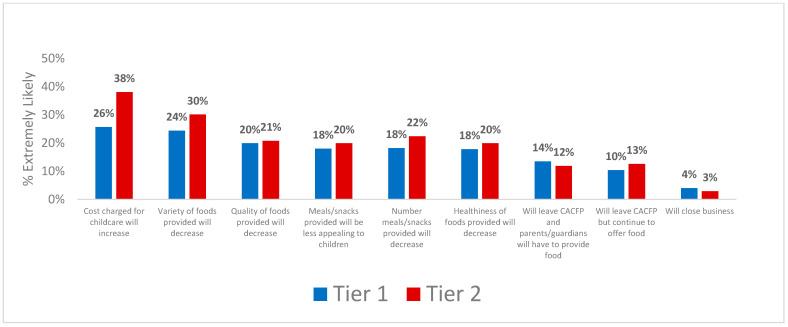
Changes anticipated by tier 1 (*n* = 261) and tier 2 (*n* = 257) family childcare home providers when the tiered Child and Adult Care Program (CACFP) reimbursements are reinstated. Response option shown is ‘extremely likely’; other response options were ‘not likely’ and ‘somewhat likely’. Significant differences between tiers were determined using logistic regression adjusting for: years licensed, years participating in CACFP, number of children in care, type of childcare offered (full, half, weekend, evening, overnight), number of meals/snacks provided daily, and provider race/ethnicity, primary language, education, number of certifications received, and household income. See Appendix A for frequencies of responses by tier).

**Table 1 nutrients-16-03241-t001:** Characteristics of the family childcare homes (FCCHs) participating in this study (*n* = 518) ^1^.

	Tier 1 (*n* = 261)	Tier 2 (*n* = 257)	*p*-Value ^2^
FCCH Provider Characteristics			
Female, *n* (%)	253 (97.3)	254 (99.2)	0.25
Age, mean (SD)	51.5 (11.0)	51.5 (10.9)	0.89
Primary language Spanish ^3^, *n* (%)	53 (20.3)	20 (7.8)	<0.001
Race/Ethnicity, *n* (%)			
Non-Hispanic Asian/Pacific Islander	19 (7.5)	54 (22.6)	<0.001
Non-Hispanic Black	32 (12.6)	16 (6.7)	0.03
Non-Hispanic Native American or American Indian	1 (0.4)	0 (0.0)	0.32
Hispanic or Latino(a)	62 (24.5)	84 (35.1)	0.03
Non-Hispanic White	131 (51.8)	67 (28.0)	<0.001
Other ^4^	8 (3.2)	18 (7.5)	0.06
Education level, *n* (%)			
High school graduate or less than high school	80 (30.7)	37 (14.6)	<0.001
Some college or Associate’s degree	122 (46.7)	113 (44.5)	0.67
Bachelor’s degree or higher	59 (22.6)	104 (40.9)	<0.001
Household income before taxes ^5^, *n* (%)			
<100% federal poverty level (FPL)	53 (21.7)	26 (11.1)	0.003
100% up to 200% FPL	141 (57.8)	156 (66.7)	0.06
≥200% up to 300% FPL	50 (20.5)	52 (22.2)	0.73
Household food secure ^6^, *n* (%)	214 (82.0)	201 (78.2)	0.55
Certifications received ^7,8^, mean (SD)	1.20 (0.64)	0.80 (0.91)	<0.001
FCCH Characteristics			
Duration in the Child and Adult Care Food Program, *n* (%)			
<1 year	6 (2.3)	11 (4.3)	0.30
1 up to 3 years	12 (4.6)	24 (9.4)	0.05
3 up to 5 years	39 (14.9)	40 (15.7)	0.90
5 up to 10 years	25 (9.6)	30 (11.8)	0.50
≥10 years	48 (18.4)	41 (16.1)	0.58
Number of children in care, mean (SD)	9.5 (4.9)	9.3 (4.3)	0.73
Number of staff, mean (SD)	2.2 (1.3)	2.2 (1.3)	0.67
Type of care offered, *n* (%)			
Full day	260 (99.6)	252 (98.1)	0.21
Half day	107 (41)	111 (43.2)	0.68
Weekends	49 (18.8)	27 (10.5)	0.01
Evenings	67 (25.7)	23 (8.9)	<0.001
Overnight	37 (14.2)	13 (5.1)	0.001
Meals/snacks provided ^8^, *n* (%)			
Breakfast	245 (94.6)	212 (86.2)	0.002
Lunch	256 (98.5)	249 (98)	0.97
Dinner/supper	178 (77.1)	131 (59)	<0.001
Morning snack	205 (84.4)	196 (86.3)	0.63
Afternoon snack	240 (95.6)	214 (91.5)	0.09
Evening snack	103 (48.8)	66 (34.4)	0.005
Number of children in care by age ^8^, Mean (SD)			
Infants (<12 months)	0.6 (1.3)	0.6 (0.9)	0.958
Toddlers (12–23 months)	1.4 (1.2)	1.8 (1.6)	0.004
Young children (2–5 years)	4.9 (3.5)	5.9 (3.8)	0.002
School-age children (6+ years)	3.0 (3.0)	1.5 (2.2)	<0.001
Primary language of children’s families, *n*%			
English	222 (85.1)	240 (93.4)	0.004
Spanish	126 (48.3)	61 (23.7)	<0.001
Chinese	11 (4.2)	40 (15.6)	<0.001
Other	17 (6.5)	35 (13.6)	0.01

^1^ Percents may add up to less than 100% due to missing data, which ranged from 0 to 7% depending on the survey question. ^2^ Significant differences between tiers were determined using *t*-test for continuous variables and chi-square for categorical variables. ^3^ Based on whether elected to complete survey in English or Spanish. ^4^ Other includes individuals who do not identify as any of the other listed categories and/or identify as one or more non-Hispanic races/ethnicities. ^5^ Based on the Federal U.S. Department of Health and Human Services Poverty Guidelines for 2023 [44]. ^6^ Assessed using the USDA 6-item food security module [45]. ^7^ Answer options included Child Development Associate (CDA), National Association for Family Child Care (NAFCC), National Association for the Education of Young Children (NAYEC), other, or none. Options were based on California Department of Education’s accepted accreditation associations [46]. ^8^ Percents may add up to more than 100% as multiple survey responses were allowed.

**Table 2 nutrients-16-03241-t002:** Estimation of the adequacy of the child and adult care food program (CACFP) reimbursement rates by family childcare home (FCCH) providers (*n* = 518) ^1^.

	Tier 1 (*n* = 261)	Tier 2 (*n* = 257)	*p*-Value ^2^
Adequacy of lower pre-COVID CACFP reimbursement rates, *n* (%) ^3^			
Not enough	179 (79.9)	194 (87.0)	0.10
Just right	43 (19.2)	28 (12.6)	0.47
Too much	2 (0.8)	1 (0.4)	0.37
Adequacy of higher COVID CACFP reimbursement rates, *n* (%)			
Not enough	156 (64.2)	105 (43.2)	<0.001
Just right	84 (34.6)	137 (56.4)	<0.001
Too much	3 (1.2)	1 (0.4)	0.24
Per child cost of serving 2 meals and 1 snack daily that meet all CACFP nutrition standards and appeal to children ages 3–5 years, *n* (%)			
≤$2.00	4 (1.6)	4 (1.6)	0.64
$2.01–$4.00	19 (7.5)	19 (7.6)	0.34
$4.01–$6.00	64 (25.1)	65 (25.9)	0.64
$6.01–$8.00	73 (28.6)	65 (25.9)	0.37
$8.01–$10.00	45 (17.6)	49 (19.5)	0.41
$10.01–$12.00	26 (10.2)	30 (12.0)	0.51
>$12.00	24 (9.4)	19 (7.6)	0.64

^1^ Counts do not add up to total tier amounts (tier 1 = 261, tier 2 = 257) due to missing data which ranged from 2 to 7% depending on the survey question. ^2^ Significant differences between tiers was determined using logistic regression adjusting for: years licensed, years participating in CACFP, number of children in care, type of childcare offered (full, half, weekend, evening, overnight), number of meals/snacks provided daily, and provider race/ethnicity, primary language, education, number of certifications received, and household income. ^3^ A higher proportion, 14%, of data were excluded due to the question not being applicable to participants or due to missing data.

## Data Availability

The data presented in this study are available within the article in Table 1 and Table 2, Figure 1, Figure 2 and Figure 3.

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
