# Peer review of "Child and Adult Care Food Program: Family Childcare Home Providers’ Perceptions of Impacts of Increased Meal and Snack Reimbursement Rates during the COVID-19 Pandemic"

_nutrients, 2024, doi:10.3390/nu16193241_

Round 1
Reviewer 1 Report
Comments and Suggestions for Authors
The manuscript presents an interesting analysis on the changes implemented in the Child and Adult Care Food Program in Californian family childcare homes, including potential impacts on diversity of foods and out-of-pocket expenditures with foods.
However, the study requires substantial changes to ensure appropriate analysis and communication of the findings, according to the following suggestions:
(1) The description of the Child and Adult Care Food Program and the family childcare homes in the Introduction should be improved, including details on type of care provided, additional information on requirements and eligibility to participate in the program (e.g., infrastructure, human resources, etc.), and potential differences in the implementation according to US state; in addition to further exploration on the literature involving effects of similar programs directed to child care in other countries
(2) The use of the term "children of color" (page 2, lines 47-48 and 51) is very inappropriate, please, see the following editorial on the subject to change the term in the study: doi:10.1001/jama.2021.13304
(3) Part of the facts stated regarding the family childcare homes participating in the Child and Adult Care Food Program lacks proper context, i.e., the periods referring to the information provided in the text regarding the proportion of FCCHs in the program, and proportion of children in the program (page 2, lines 74-76)
(4) The term natural experiment should not be presented using quotation marks, considering that comprises a definition usually used in the literature
(5) A parenthesis is missing on the description presented in line 109 (page 3)
(6) Questions presented in the survey focus on perceptions of the individuals interviewed, therefore, the measures examined in the study should be defined as "perceived benefits", "perceived impact", "estimated cost", "perceived ability" (page 2, lines 117-124)
(7) Several questions (17 to 21) in the survey required responses using 3-point or 4-point Likert scales (some including items that should be considered "missing"), which could be included in the description of the measures (page 3, lines 123-128); in addition, these responses should be presented and analyzed using frequencies instead of mean and standard deviation
(8) Furthermore, it is very odd to use linear and logistic regressions to establish differences between groups, especially considering that meal costs (example of continuous variable) was presented in options referring to intervals of costs (therefore, should be assessed as frequency of individuals choosing each specific option), and chi-square is the recommended test for identification of differences in frequencies between groups
(9) The type of care offered (presented in Table 1) should be described in details in the Introduction to provide context for readers
(10) The results in Figures 1, 2 and 3 should be presented in frequencies of individuals choosing each specific option, and differences in responses according to groups (tier 1 and 2) should be performed using chi-square test, in addition to indicating that represent perceptions of respondents (pages 6-8)
Finally, the Discussion should encompass an in-depth exploration of the findings of the study, comparing with evidence in the literature regarding effects of similar programs in the US and other countries (i.e., beyond the studies focusing on the analysis of Child and Adult Care Food Program). The current configuration of the section presents a limited approach of the subject and implications of the findings on potential changes that could be proposed for the program.
Author Response
Reviewer comment 0: The manuscript presents an interesting analysis on the changes implemented in the Child and Adult Care Food Program in Californian family childcare homes, including potential impacts on diversity of foods and out-of-pocket expenditures with foods. However, the study requires substantial changes to ensure appropriate analysis and communication of the findings, according to the following suggestions:
(1) The description of the Child and Adult Care Food Program and the family childcare homes in the Introduction should be improved, including details on type of care provided, additional information on requirements and eligibility to participate in the program (e.g., infrastructure, human resources, etc.), and potential differences in the implementation according to US state; in addition to further exploration on the literature involving effects of similar programs directed to child care in other countries.
Author response: Thank you for your helpful comments. As requested we have added more information about licensed childcare in the U.S. in the introduction (page 1, lines 37-43 and page 2, lines 78-85) . Exploration of some literature in other countries has been added to the discussion (page 10, lines 345-353). Additional emphasis has also been added to the discussion that this is a study conducted in the U.S. (e.g., page 9, lines 298, 308, 312)
(2) The use of the term "children of color" (page 2, lines 47-48 and 51) is very inappropriate, please, see the following editorial on the subject to change the term in the study: doi:10.1001/jama.2021.13304
Author response: Thank you for this reminder. We have updated the text accordingly (page 2, lines 56-62).
(3) Part of the facts stated regarding the family childcare homes participating in the Child and Adult Care Food Program lacks proper context, i.e., the periods referring to the information provided in the text regarding the proportion of FCCHs in the program, and proportion of children in the program (page 2, lines 74-76)
Author response: We have added time periods for all factual statements and updated a reference to cite national averages to be more consistent with the sentence that follows (page 2 lines 91-92, 97-98; page 3, lines 99-101).
(4) The term natural experiment should not be presented using quotation marks, considering that comprises a definition usually used in the literature
Author response: The quotation marks have been removed (page 3, lines 101-102).
(5) A parenthesis is missing on the description presented in line 109 (page 3)
Author response: The missing parenthesis has been added (page 3, line 126).
(6) Questions presented in the survey focus on perceptions of the individuals interviewed, therefore, the measures examined in the study should be defined as "perceived benefits", "perceived impact", "estimated cost", "perceived ability" (page 2, lines 117-124)
Author response: We have added ‘perceived’ and ‘estimated’ as suggested (page 2, lines 134-138 and elsewhere as relevant).
(7) Several questions (17 to 21) in the survey required responses using 3-point or 4-point Likert scales (some including items that should be considered "missing"), which could be included in the description of the measures (page 3, lines 123-128); in addition, these responses should be presented and analyzed using frequencies instead of mean and standard deviation
Author response: We have added clarification that missing responses were not scored (page 3 lines 142-146) and that means/SDs were computed for scored variables (page 4, line 149). With survey questions involving ordinal or Likert-scale responses, it is acceptable to score them and present means and standard deviations (see discussion here). This is routinely done when examining multiple variables using a Likert scale and to improve the ability to detect significant differences as well as facilitate interpretation. We therefore have not changed the way the data are presented, but have added two supplemental tables so that readers can examine the frequencies and the scored means/SDs from the relevant figures. Supplemental table 1 corresponds to Figure 2. Supplemental Table 2 corresponds to Figure 3.
(8) Furthermore, it is very odd to use linear and logistic regressions to establish differences between groups, especially considering that meal costs (example of continuous variable) was presented in options referring to intervals of costs (therefore, should be assessed as frequency of individuals choosing each specific option), and chi-square is the recommended test for identification of differences in frequencies between groups
Author response:
A chi-square is a special case of unadjusted logistic regression where both the independent and dependent variables are dichotomous. Because chi-square models are bivariate (and do not allow for the control of potential confounders), we are using logistic regression to include multiple continuous and categorical potential confounders as predictor variables. Adjusting for confounders is the standard (best) practice for analyzing data from observational studies
(9) The type of care offered (presented in Table 1) should be described in detail in the Introduction to provide context for readers
Author response: We have added more detail in the introduction to describe that family childcare homes can operate under a range of conditions (please see comment 1 above).
(10) The results in Figures 1, 2 and 3 should be presented in frequencies of individuals choosing each specific option, and differences in responses according to groups (tier 1 and 2) should be performed using chi-square test, in addition to indicating that represent perceptions of respondents (pages 6-8)
Author response: We have added to the figures that the results represent perceptions of respondents. See comments 7 and 8 above for a justification for using logistic regression instead of chi-square.
Finally, the Discussion should encompass an in-depth exploration of the findings of the study, comparing with evidence in the literature regarding effects of similar programs in the US and other countries (i.e., beyond the studies focusing on the analysis of Child and Adult Care Food Program). The current configuration of the section presents a limited approach of the subject and implications of the findings on potential changes that could be proposed for the program.
Author response: The change to the CACFP reimbursement rates was a response to the COVID-19 pandemic and therefore has not been done previously in the US. We examined the literature in other countries as requested (page 10, lines 346-354).
Reviewer 2 Report
Comments and Suggestions for Authors
Higher tier 1 reimbursements are for providers who operate in low-income communities or who have low income themselves (Complete this thought, I think the researchers meant who ARE low income themselves)
Over half reported lowering out-of-pocket spending for food (59%) and serving greater variety (55%) and quality (54%) of 23 foods. (awkward sentence, does this sentence include both tier 1 and 2 programs. Need to word this statement well and also add more information)
In Spring 2023, a sample of 2,000 licensed California FCCHs who participated in 90 CACFP (half tier 1; half tier 2) was randomly selected from a total of approximately 28,000 91 FCCHs in state administrative data.(it should be were selected)
I would like the researchers explicitly state research questions that they are examining in this paper
Comments on the Quality of English LanguageHigher tier 1 reimbursements are for providers who operate in low-income communities or who have low income themselves (Complete this thought, I think the researchers meant who ARE low income themselves)
Over half reported lowering out-of-pocket spending for food (59%) and serving greater variety (55%) and quality (54%) of 23 foods. (awkward sentence, does this sentence include both tier 1 and 2 programs. Need to word this statement well and also add more information)
In Spring 2023, a sample of 2,000 licensed California FCCHs who participated in 90 CACFP (half tier 1; half tier 2) was randomly selected from a total of approximately 28,000 91 FCCHs in state administrative data.(it should be were selected)
I would like the researchers explicitly state research questions that they are examining in this paper
Author Response
Higher tier 1 reimbursements are for providers who operate in low-income communities or who have low income themselves (Complete this thought, I think the researchers meant who ARE low income themselves)
Author response: Thank you for your helpful comments. We had selected to use the language ‘living on a low income’ rather than ‘are low income’ to use more inclusive languages (See support for this practice here) (page 1, lines 17-18).
Over half reported lowering out-of-pocket spending for food (59%) and serving greater variety (55%) and quality (54%) of 23 foods. (awkward sentence, does this sentence include both tier 1 and 2 programs. Need to word this statement well and also add more information)
Author response: In response to your question, as stated on page 6 lines 205-206, these results include both tier 1 and tier 2 combined. We have added clarification to better ensure that readers understand this in the results (page 7, line 227) and in the abstract (page 1, lines 22-23)
In Spring 2023, a sample of 2,000 licensed California FCCHs who participated in CACFP (half tier 1; half tier 2) was randomly selected from a total of approximately 28,000 FCCHs in state administrative data.(it should be were selected)
Author response: Because the noun in this sentence is ‘a sample’ (singular), ‘was’ is technically correct. However, we agree that this sentence sounds awkward and have changed it by deleting ‘a sample’ so that the sentence now reads ‘In Spring 2023, 2,000 licensed California FCCHs who participated in CACFP (half tier 1; half tier 2) were randomly selected (page 3, lines 108-109).
I would like the researchers explicitly state research questions that they are examining in this paper
Author response: We have reframed our study aim (originally stated as: Our study aimed to assess the impacts of the increased CACFP reimbursements on the benefits of participating in the program and the potential challenges anticipated with reinstating the tiered reimbursement rates as reported by FCCH providers in California) as research questions (page 3, lines 102-105).
Round 2
Reviewer 1 Report
Comments and Suggestions for Authors
Authors made changes referring to major part of my suggestions; however, the main suggestion regarding the use of frequencies to report responses of individuals was ignored. The presentation of results using mean and standard deviation of responses based on Likert scales is wrong, considering that scores within Likert scales comprise ordinal levels of measurement, please see:
https://www.jstor.org/stable/1388904
doi:10.1097/00000637-199609000-00006 (sin #1)
doi:10.1162/pres.16.4.439
doi:10.1111/j.1365-2929.2004.02012.x
Therefore, the analyses performed present high risk of bias.
Author Response
Authors made changes referring to major part of my suggestions; however, the main suggestion regarding the use of frequencies to report responses of individuals was ignored. The presentation of results using mean and standard deviation of responses based on Likert scales is wrong, considering that scores within Likert scales comprise ordinal levels of measurement, please see:
https://www.jstor.org/stable/1388904
doi:10.1097/00000637-199609000-00006 (sin #1)
doi:10.1162/pres.16.4.439
doi:10.1111/j.1365-2929.2004.02012.
Therefore, the analyses performed present high risk of bias.
We appreciate the reviewer's expertise on this (although our epidemiologist has provided us with papers that argue that our original analysis is an acceptable option). Therefore, we have made all of the requested changes: frequencies are now reported in Figures 2 and 3 instead of means/SDs and logistic regression is used instead of linear regression. This did not result in any changes to statistical significance or our conclusions. Our revisions are marked in blue - to distinguish them from the initial revisions marked in red.